# All-Dielectric Structural Colors with Lithium Niobate Nanodisk Metasurface Resonators

**Yuting Zhou** *[ID]**, Qingyu Wang, Zhiqiang Ji and Pei Zeng**

College of Mechanical and Vehicle Engineering, Hunan University, Changsha 410082, China;
wangqingyu@hnu.edu.cn (Q.W.); zhiqiangji@hnu.edu.cn (Z.J.); zengpei@hnu.edu.cn (P.Z.)
* Correspondence: zzyytt@hnu.edu.cn

**Abstract:** Lithium niobate (LN) is a promising optical material, its micro–nano structures have been applied to fields such as photonic crystals, nonlinear optics, optical waveguides, and so on. At present, lithium niobate structural colors are rarely studied. Although the nanograting structure was researched, it has such large full width at half-maximum (fwhm) that it cannot achieve red, green, or blue pixels or other high-saturation structural colors, thus, its color printing quality is poor. In this paper, we design and simulate lithium niobate nanodisk metasurface resonators (LNNDMRs), which are based on Mie magnetic dipole (MD) and electric dipole (ED) resonances. In addition, the resonators yield very narrow reflection peaks and high reflection efficiencies with over 80%, especially the reflection peaks of red, green, and blue pixels with fwhm around 11 nm, 9 nm, and 6 nm, respectively. Moreover, output colors of different array cells composed of single nanodisk in finite size are displayed, which provides a theoretical basis for their practical applications. Therefore, LNNDMRs pave the way for high-efficiency, compact photonic display devices based on lithium niobate.

**Keywords:** lithium niobate; nanodisk metasurface resonators; Mie resonance; structural colors; array cells

## 1. Introduction

In nature, the wings of Morpho butterflies [1,2] and the feathers of peacocks [3,4] display vivid colors, known as "structural colors", which are the result of the interaction of visible light with periodic nanostructured materials on their surfaces. Compared to conventional dye pigments, which are susceptible to optical damage and somewhat hazardous to the environment [5], structural-color-based pixels offer the merits of high resolution, good sustainability, and environmental friendliness. Thus, structural colors have become a fascinating field in science and engineering in recent years [1,6–9]. For artificial structural colors, methods based on Fabry−Perot (FP) cavity multilayer structure [10,11] and plasmonic nanostructures [12,13] have emerged in large numbers; however, there are still some challenges. Typically, the practical application of multilayer structures is still limited due to complex fabrication processes and durability and oxidation issues [14]. In addition, for plasmonic nanostructures, their reflection peaks are generally broad and less intensive [12,15] due to the inherent loss of metals in the visible region. Furthermore, although the approach of applying the metasurface concept to plasmonic metals boosts light–matter couplings [16–18], the intrinsic Ohmic losses still hinder their applications.

To solve these problems, all-dielectric metasurface materials with high refractive indices have become brilliant alternatives. Proust et al. [19] utilized Si Mie resonators and achieved an all-dielectric color metasurface, but silicon has very strong absorption in the visible light range (when the wavelength λ is smaller than 450 nm, the imaginary part of permittivity sharply increases with the decrease of λ), which directly brings about challenges in generating blue and purple pixels. Xiao et al. [20] reported on the generation

of colors in $TiO_2$ metasurfaces, the cross-section of the structure is a 72° trapezoidal face that is difficult to control and fabricate. Recently, due to several advantages of lithium niobate ($LiNbO_3$, LN) (i.e., its large refractive index, wide transparency wavelength, and large second-order nonlinear coefficient), it has been widely studied and applied in the fields of optics and photonics [21–24]. A complete set of optical components has been developed on the LN platform with decent performance, such as nonlinear wavelength converters [25] and broadband frequency comb sources [26], as well as photon-pair sources [27]. The functional devices require the fabrication of micro–nano structures. Due to its stable physical and chemical characteristics, it is a challenge for chemical etching and mechanical sculpture. Fortunately, until now, laser beam writing [28], focused ion beam milling [29] and other fabrications [30,31], has emerged for the processing of lithium niobate micro and nanostructures. Moreover, a technique called "smart cut" [32,33] has been developed specifically for the production of various thin-film materials. Through a series of processes such as ion implantation, direct bonding, and thermal annealing, the technique can peel off a single-layer lithium niobate single-crystal film from the bulk material and transfer it to the target substrate, and its properties are almost the same as the bulk material performances. Based on the above analysis and processing methods, however, structural colors based on lithium niobate are rarely reported, even if only the nanograting structure of LN on quartz was researched to generate transmission colors [34], but the full width at half-maximum (fwhm) of the nanostructure is large, resulting in a low saturation, which means that other spectral components corresponding to other colors are incorporated. Thus, the lithium niobate nanograting metasurfaces are incapable of generating red, green, and blue (RGB) color pixels and then causing low-quality structural color printing.

Here, we design and numerically simulate lithium niobate nanodisk metasurface resonators (LNNDMRs) and find that the resonators exhibit narrow peaks and high reflection efficiencies, meanwhile, we demonstrate that electric and magnetic dipole moments are well-formed in lithium niobate nanodisk metasurface resonators. Additionally, LNNDMRs shows output colors of different array cells consisting of single nanodisks in finite size.

## 2. Results and Discussion

Figure 1a exhibits the structure of all-dielectric lithium niobate nanodisk metasurface resonators (LNNDMRs) with the multi-functional material lithium niobate (LN) as the metasurface structure and low refractive index glass ($SiO_2$) as the substrate. The period (P) and diameter (D) are tunable parameters that allow the resonance peaks to shift in the visible wavelength range. The optical constants of LN and glass are plotted in Figure 1b. In order to design metasurface resonators with better regularity, we define the period occupancy rate (POR) of each structure, that is, the radius divided by the period; the expression is POR = (D/2/P) × 100%. Figure 1c shows the reflection spectra of the RGB color pixels based on LN nanodisk resonators, the fwhm of red, green, and blue reflection spectra are around 11 nm, 9 nm, and 6 nm, respectively. Lithium niobate nanodisk metasurface resonators (LNNDMRs) generate the very narrow fwhm, resulting in higher saturation colors than structures in published papers [35]. In addition, the dimensions of the structures are also listed in the legend; thus, we can conclude that the PORs of RGB nanodisk metasurface resonators are around 27.25%, 27.03%, and 25%, respectively. The corresponding output colors of the RGB color pixels are depicted in a standard International Commission on Illumination (CIE) 1931 chromaticity diagram (Figure 1d), indicating the generation of the high-saturation RGB colors by virtue of the proposed metasurfaces with the specially selected periods and diameters.

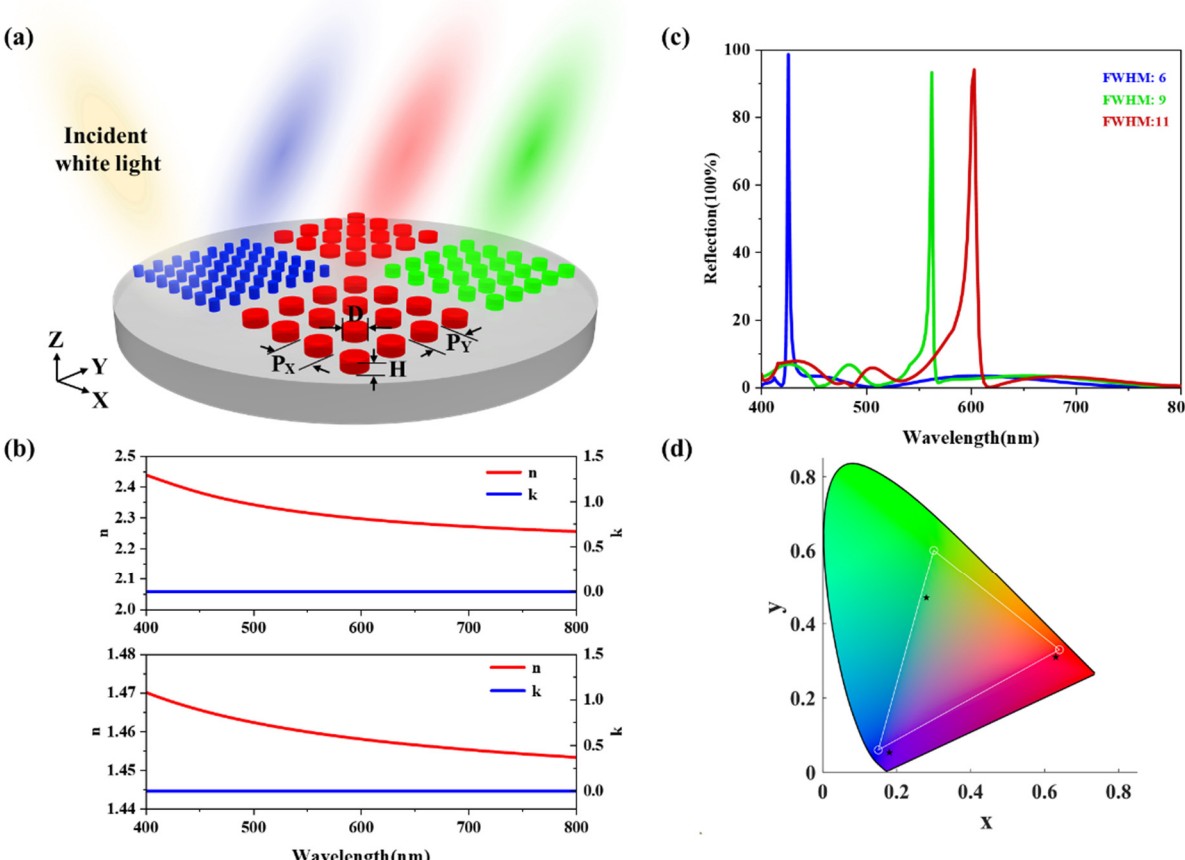

**Figure 1.** Reflection spectra and structural colors. (**a**) Schematic of the arrays of LN nanodisks on a glass substrate. The height of the nanodisks is 500 nm (H = 500 nm); the diameter (D) and the period (Px = Py = P) are free parameters. (**b**) The top shows optical constants of LN; the bottom shows optical constants of glass. The refractive index (n) and extinction coefficient (k) are illustrated as red and blue solid lines, respectively. (**c**) Reflection spectra of the RGB color pixels based on LNNDMRs. Dimensions: D = 218 nm, P = 400 nm (red); D = 200 nm, P = 370 nm (green); D = 140 nm, P = 280 nm (blue). (**d**) The direct output (black pentagram) structural colors from reflection spectra of LNNDMRs with RGB in CIE 1931 color map.

After that, based on the structures of the RGB pixels, the positions of the reflection peaks are only shifted by changing the diameter, which results in the generation of high saturation structural colors. Figure 2a shows several output colors with diameters from 132 nm to 148 nm when the period is 280 nm. As the diameter increases, the colors change from purple to blue. Figure 2b shows the output colors changing with the diameter when the period is 370 nm; the structure obtains colors such as green, chartreuse, dark green, and so on. Figure 2c shows the situation of the period at 400 nm; red, pink, and wine colors are displayed. Thus, LN nanodisk metasurface resonators not only produce RGB pixels and other brilliant colors, but also possess high saturations compared to previously reported lithium niobate nanograting metasurfaces [34]. Likewise, we can conclude that, with the change of diameter, the reflections of LNNDMRs are almost above 80% with narrow full widths at half-maximum. Upon further consideration, such brilliant output colors are manifested due to the higher refractive index contrast between the all-dielectric lithium niobate and the glass. Mie resonance theories explain the phenomenon [36]. In principle, each lithium niobate nanodisk resonator sustains a Mie electric dipole (ED) resonance and a Mie magnetic dipole (MD) resonance. We design periodic nanodisks so that the proximity resonance can occur when the Mie multiple resonators are arrayed closely, effectively enhancing the magnitude of the resonant modes. Therefore, lithium niobate nanodisk metasurface resonators (LNNDMRs) are formed. Under further analysis, the periodically

placed lithium niobate nanodisks can even form the so-called photonic band gap, which shall suppress other high-order resonant modes and remarkably improve obvious color impressions.

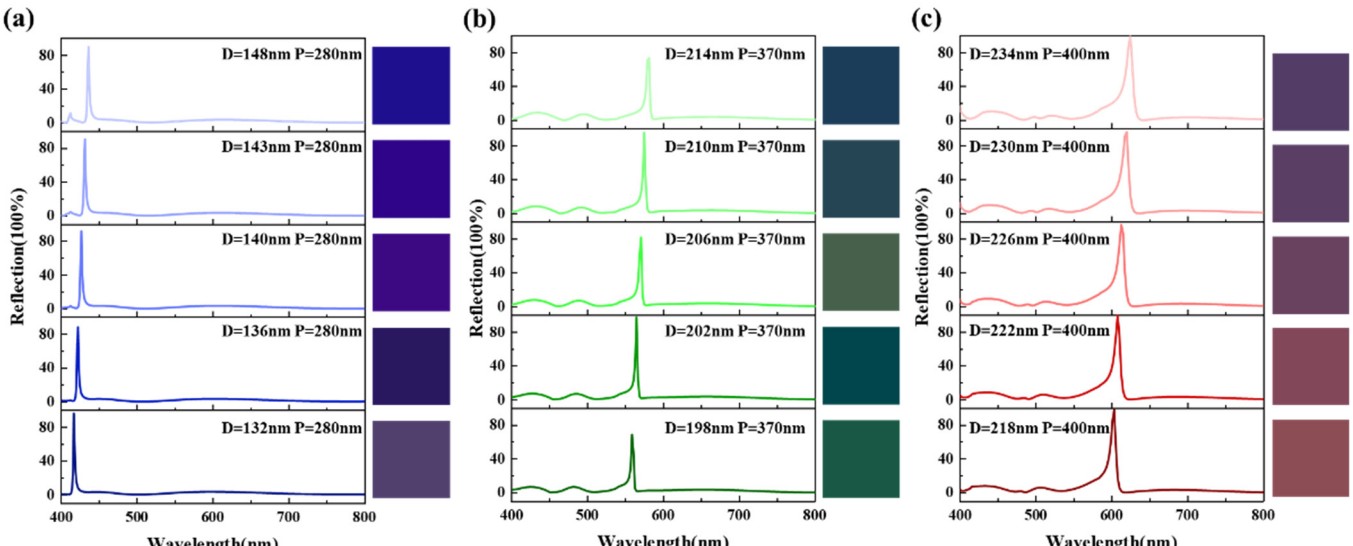

**Figure 2.** Simulated reflection spectra and structural colors. (**a**) Output colors for LNNDMRs at a period of 280 nm. (**b**) Output colors for LNNDMRs at a period of 370 nm. (**c**) Output colors for LNNDMRs at a period of 400 nm.

For the above analysis, we numerically simulate and calculate the reflection spectra of LNNDMRs as a function of nanodisk diameter (Figure 3b) with commercial finite-difference time–domain software (Lumerical FDTD Solutions 2018a). The incident light is along the z-axis, and the polarization direction is along the x-axis (Figure 3a). When the diameter is 206 nm, the maximum reflection peak appears at the wavelength of 587 nm. With the increase of diameter, the resonant mode continuously shifts to longer wavelengths. To further verify the mechanism of the resonance in LNNDMRs, we select two representative resonance cases at a diameter of 218 nm and mark them with yellow pentagrams, labeled Mode i and Mode ii, respectively (Figure 3b). Simultaneously, Figure 3c depicts electric (E) field distributions at the above two resonant peaks, in which the upper group of diagrams show the field distribution on the z = 0 plane, and the lower group of diagrams show the field distribution on the y = 0 plane. For the resonance Mode i at 605 nm, a resonance peak is excited, which is due to the magnetic dipole (MD) mode. In addition, the conclusion can be proved by the electric field distribution from the y = 0 plane (Figure 3c-Mode i), there is a clear circulating electric field loop inside the LN nanodisks, which partially extends into the glass. Moreover, the MD resonance-enabled reflection peak tends to show redshift, which has been explained in the literature, and the redshift of the MD resonance requires the analysis of the physics behind the photo-induced processes taking place in the nanodisks [37]. Similarly, we can conclude that the obtained narrow band reflection spectrum of the proposed red pixel, as presented in Figure 2c, shows simple resonance. While, for the resonance Mode ii at 700 nm, the resonance shows a more complex electric dipole (ED) mode with two adjacent circulating electric currents in the incidence plane and weak electric field enhancement (Figure 3c-Mode ii), which is in good agreement with previous works [38–40]. Meanwhile, in Figure 3b, since the magnetic resonance is the fundamental resonant mode, there are no resonances on the red peak side. In addition, the periodic nanodisk yields a band gap that beneficially suppresses the higher-order modes. In this sense, the lithium niobate nanodisk metasurface resonators (LNNDMRs) can have notable advantages in displaying structural colors, such as saturation and contrast.

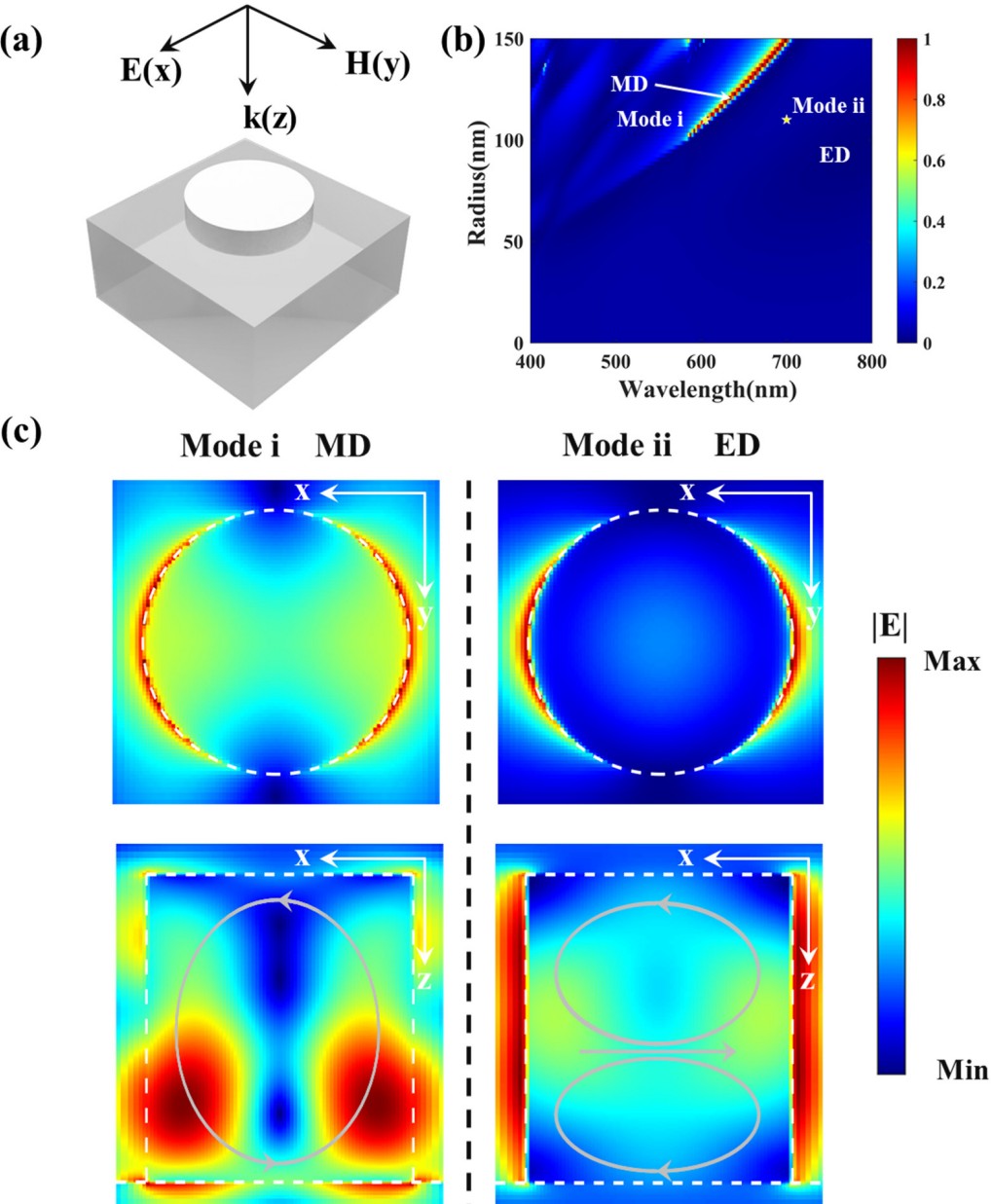

**Figure 3.** Mechanism of LNNDMRs. (**a**) Schematic illustration of a LN nanodisk resonator with plane wave excitation, where the polarization is along the x-axis; k is the wave vector of the incident white light. The origin of the coordinate system is placed at the center of the structure. (**b**) The reflection spectra versus different radiuses of LN nanodisks with periodic P fixed to 400 nm. Yellow pentagrams refer to the chosen two resonance wavelengths corresponding to a diameter of 220 nm. (**c**) E-field distributions at resonances Mode i and Mode ii on the y = 0 and z = 0 plane. The structure of LNNDMRs, comprising LN nanodisk and glass substrate, is denoted by the white dashed line. The grey arrow lines give a schematic representation of the instantaneous electric field lines around the resonator.

On the other hand, for the above output colors, the simulations are performed under the periodic boundary conditions of the lithium niobate nanodisk metasurface resonators (LNNDMRs), that is, infinite arrays are assumed to extend in the x-direction and y-direction. Meanwhile, the infinite arrays of nanodisks are impossible to fabricate, and the color pixels display structural colors at a finite size [41]. Therefore, considering the possibility in practical applications, we simulate the output colors of different array cells consisting of single nanodisks with the same diameter and period, which provides a theoretical basis for industrial production. Figure 4 depicts the output colors of a variety of array cells. Perfect matching layer (PML) conditions are applied along the x-direction, y-direction, and z-direction in the commercial software of Lumerical FDTD Solutions 2018a. The results obtained for the array cells consisting of a single nanodisk with a diameter of 218 nm and a period of 400 nm are shown in Figure 4I. The reflection spectra of the array cells of $4 \times 4$, $8 \times 8$, $16 \times 16$, and $32 \times 32$ are calculated in Figure 4I-a, respectively. Simultaneously, Figure 4I-b plots the changes to the reflection peak and full width at half-maximum in the case of the different array cells. With the increase of the number of single nanodisks in the array cells on LNNDMRs, the reflection peak becomes stronger at full width and at half-maximum (fwhm) becomes narrower. The result proves the previously mentioned theory that periodic densely placed nanodisk array cells effectively increase the amplitude of the resonant mode and create a so-called photonic band gap, which shall narrow fwhm and inhibit other high-order resonant modes. On the basis of reflection spectra, colors are calculated and shown in Figure 4I-c. The color of array cells at the dimension on LNNDMRs gradually tends to red, in particular, at $32 \times 32$ array cells, the reflection peak is more than 50%, and the FWHM is also below 20 nm. To further ensure the ability of LNNDMRs to generate other colors, the consequence of array cells consisting of a single nanodisk with D = 202 nm, P = 370 nm and D = 140 nm, P = 280 nm are shown in Figure 4II,III, respectively. Similarly, for the nanodisk structure with dimensions D = 202 nm and P = 370 nm, the reflection peak is close to 50%, the FWHM is also lower than 20 nm (Figure 4II-d,e), and the color is prone to develop towards green at $32 \times 32$ array cells. However, for the size of D = 140 nm, P = 280 nm, the reflection peak is close to 35% at $46 \times 46$ array cells (Figure 4III-g,h), and the LNNDMRs gradually show blue (Figure 4III-i). We believe that the results are related to the period occupancy rate (POR) of each structure (as defined above), because the POR of the structure in Figure 4III is smaller than that of the other two structures. We obtain structural colors at finite sizes, demonstrating that LNNDMRs are helpful and effective.

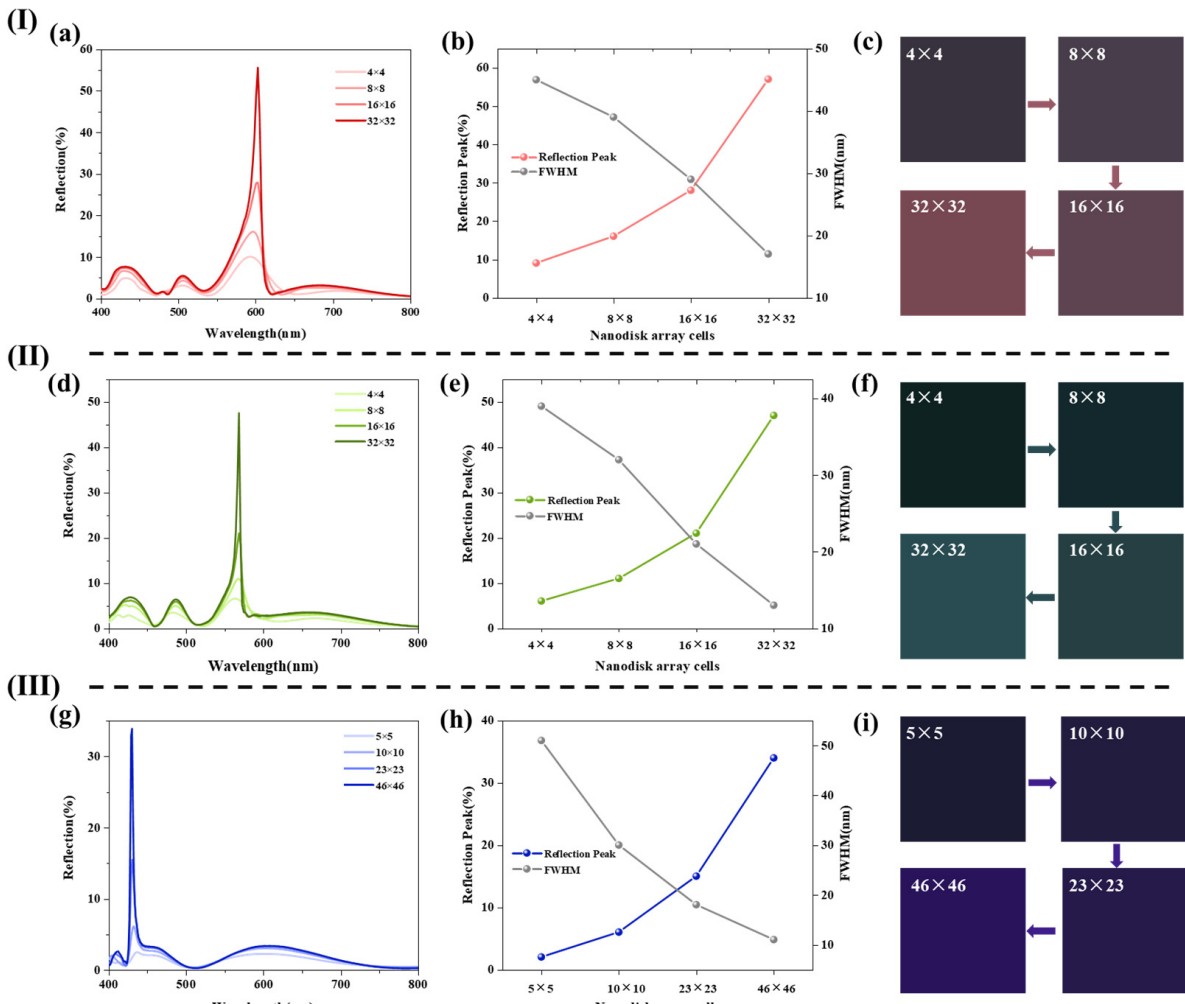

**Figure 4.** Nanodisk array cells of LNNDMRs. (**I**) Dimensions: D = 218 nm, P = 400 nm: (**a**) Reflection spectrum at 4 × 4, 8 × 8, 16 × 16, and 32 × 32 array cells. (**b**) Reflection peaks and FWHM for different array cells. (**c**) Output colors with different array cells; (**II**) Dimensions: D = 202 nm, P = 370 nm: (**d**) Reflection spectrum at 4 × 4, 8 × 8, 16 × 16, and 32 × 32 array cells. (**e**) Reflection peaks and FWHM for different array cells. (**f**) Output colors with different array cells; (**III**) Dimensions: D = 140 nm, P = 280 nm: (**g**) Reflection spectrum at 5 × 5, 10 × 10, 23 × 23, and 46 × 46 array cells. (**h**) Reflection peaks and FWHM for different array cells. (**i**) Output colors with different array cells.

## 3. Conclusions

In summary, we designed an all-dielectric lithium niobate nanodisk metasurface resonators (LNNDMRs) to determine structural colors. Through numerical simulation and analysis, the resonators generate very narrow and high-intensity reflection peaks, and output colors of the RGB pixels are plotted in the CIE color map. Meanwhile, chartreuse, pink, purple, wine color, and other output colors are obtained with the change of diameter in the case of different periods, proving that the structure of the LNNDMRs produces abundant structural colors. For this phenomenon, by analyzing the electric (E) field distribution of the nanodisk on the y = 0 plane, it is explained that the reflection peaks of lithium niobate nanodisk metasurface resonators are excited by Mie magnetic dipole and electric dipole resonances. Moreover, we simulated the output colors of different array cells composed of single nanodisks in finite size, which provides a theoretical basis for production. Therefore, it has wide prospects in the practical application of high-efficiency and compact display devices based on lithium niobate.

**Author Contributions:** Conceptualization, Y.Z.; methodology, Y.Z.; software, Y.Z.; validation, Y.Z., Q.W. and Z.J.; formal analysis, Y.Z. and P.Z.; investigation, Y.Z., Q.W., Z.J. and P.Z.; resources, Y.Z., Q.W., Z.J. and P.Z.; data curation, Y.Z., Q.W., Z.J. and P.Z.; writing–original draft preparation, Y.Z.; writing–review and editing, Y.Z., Q.W., Z.J. and P.Z.; visualization, Y.Z.; supervision, Y.Z.; project administration, Y.Z. All authors have read and agreed to the published version of the manuscript.

**Funding:** This research received no external funding.

**Institutional Review Board Statement:** Not applicable.

**Informed Consent Statement:** Not applicable.

**Data Availability Statement:** Not applicable.

**Acknowledgments:** The authors thank Hunan University for its help and support. The authors would like to thank the editors and anonymous reviewers for giving valuable suggestions that have helped to improve the quality of the manuscript.

**Conflicts of Interest:** The authors declare no conflict of interest.

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
