# Peer review of "All-Dielectric Structural Colors with Lithium Niobate Nanodisk Metasurface Resonators"

_photonics, doi:10.3390/photonics9060402_

Round 1

Reviewer 1 Report

The authors report a numerical investigation on Lithiun Niobate metasurface for generating structural color. The paper is interesting, but the following issue need to be address before pubblicaiton.

It is necessary to better clarify why the color present a little dependence on incident angle.

Which is the collection angle? Do these metasurface present a dependence on collection angle?

It is a missing opportunity to not cite manuscript related to structural colors formation with multilayer design which are very easy to be fabricated. As an example:

  • Nature Mater 12, 20–24 (2013)
  • Optical Mater. 2020, 8, 2000487
  • ACS Photonics 2019, 6, 9, 2342–2349

Why the author introduced period occupancy and not use just fill factor as parameter?

In line 75 the dimension of nanostructure should be reported. How they found these parameters?

It not a good way to indicate resonance with “i” and “ii”

English need to be improved because it is not very easy to read the manuscript.

Author Response

Dear reviewer:

Thank you very much for your valuable comments and suggestions. We have revised the paper thoroughly following the comments and questions raised. Indeed, these changes significantly improved the quality of the paper. We hope that the revised paper now meets the high standard of the journal. All the changes have been highlighted in the revised manuscript. English of the paper has been substantially modified from co-authors with a good command of English, and these are not all marked.

We have uploaded the response letter and revised manuscript.

Thank you for your review again, and we look forward to hearing from you.

We wish you luck and happiness everyday!

Author Response

(The authors gave the same response as above.)

Reviewer 3 Report

Comments on the manuscript

In this manuscript entitled “All-Dielectric Structural Colors with Lithium Niobate Nano-disk Metasurface Resonators” the authors introduced the simulation results of ultra-narrow resonances supported in lithium niobate metasurfaces for structural color manipulation. Their studies employed Mie magnetic dipole (MD) and electric dipole (ED) resonances to boost the Q-factor and the transmission efficiency of resonances. They obtained high Q Mie resonance for red, green, and blue colors by changing the diameters and the period of a unit cell of metasurfaces. Although the topic of Mie resonances for structural color is not new, the manuscript is technically sound with conclusions and assertions well-supported by numerical results. However, this manuscript did not provide deep insight into physics but instead presented superficial results. Also, the language is a little difficult to understand, especially in the main text section, which degrades the whole level of the manuscript.

Thus, this manuscript needs substantial revision before it can meet the scope of Photonics. Consequently, there are a few points that should be addressed for the purpose of clarity.

  1. My major concern is the selling point of the manuscript. The authors did present interesting simulation results with high-Q Mie resonance in lithium niobate (LN) metasurfaces, which is nice and attractive to the readers. The authors also argue that there are rare reports on LN metasurfaces for structural color. This is correct. However, in practical fabrication, the geometrical shape of the LN nanodisk is limited by non-ideal etching and thus it is difficult to guarantee disk shape for LN particles. That is the reason why researchers turned to easy etch materials such as Si and TiO2 rather than LN. Have the authors ever thought about the difficulties of their design in practical implementation?
  2. The simulations were based on periodical conditions for the LN metasurfaces, which assume infinite arrays extending in x- and y-direction. Also, the high Q-factor Mie resonance arrays are based on a non-local effect where the collective resonances play a crucial role. However, in practical application, the color pixel is often of finite size (say 3*3 or 5*5unit cells). Have the authors thought about the finite-size effects?
  3. “Another significant advantage of lithium niobate nanodisk metasurface resonators (LNNDMRs) is the especially weak angular dependence.” As for the ultra-weak angular dispersion effects shown in Figure 4. I have concerns about the validity of the authors' simulation results and methods. In Figure 4a-4d, it appears that all the resonance peaks, no matter strong or weak, are weakly dependent on the oblique incidence, suggesting a negligible influence of the Rayleigh anomalies/higher-order diffractions. This is non-physics. Nano-disk, nano-hole, and nano-grating arrays are often of strong angular dispersion effects for most of the resonance modes [please see “Hsu, Chia Wei, Bo Zhen, Jeongwon Lee, Song-Liang Chua, Steven G. Johnson, John D. Joannopoulos, and Marin Soljačić. "Observation of trapped light within the radiation continuum." Nature 499, no. 7457 (2013): 188-191”; “Klopfer, Elissa, Sahil Dagli, David Barton III, Mark Lawrence, and Jennifer A. Dionne. "High-Quality-Factor Silicon-on-Lithium Niobate Metasurfaces for Electro-optically Reconfigurable Wavefront Shaping." Nano Letters 22, no. 4 (2022): 1703-1709”]. What caused the difference between the authors’ results and those obtained by others? Have the authors used the Broadband Fixed Angle Source (BFAST) technique in their simulations? Please see the supporting information in [Liang, Yao, Han Lin, Shirong Lin, Jiayang Wu, Weibai Li, Fei Meng, Yunyi Yang, Xiaodong Huang, Baohua Jia, and Yuri Kivshar. "Hybrid anisotropic plasmonic metasurfaces with multiple resonances of focused light beams." Nano Letters 21, no. 20 (2021): 8917-8923]. I would recommend double-checking the simulation results.
  4. I agree that the ultra-weak angular dispersion is an attractive feature for metasurfaces with high Q resonances. This aspect needs to be highlighted in the manuscript. The third paper in my comment 3 many help the authors.

Author Response

(The authors gave the same response as above.)

Reviewer 4 Report

In this manuscript, the authors designed an all-dielectric structural colors of lithium niobate nanodisk metasurface resonators. Through numerical simulation and analysis, it is found that the resonators generate very narrow and high-intensity reflection peaks, and output colors of the RGB pixels are plotted in the CIE color map. Meanwhile, chartreuse, pink, purple, wine color, and other output colors are obtained with the change of diameter in the case of different periods, proving that the structure of LNNDMRs produces abundant structural colors. In addition, the nanodisk metasurface resonantors exhibit a quite weak angle dependence at the incident angle from 0o to 70o, and the resonance peaks cannot occur redshift or blueshift, only the reflection intensity changes.

The reviewer has the following comments and suggestions:

1. Besides the nanodisk sturtcure, can we still get the same or similar results with other nanostrutcures, e.g., nanocube and nanorod?

2. "Further analysis, the periodically placed lithium niobate nanodisks can even form the so-called photonic band gap, which shall suppress other high order resonant modes and remarkably improve obvious color impressions.”.

“And the periodic nanodisk yields a band gap that beneficially suppresses the higher order modes. In this sense, lithium niobate nanodisk metasurface resonators (LNNDMRs) can have notable advantages in displaying structural colors, such as saturation and contrast.”

Could the authors give the photonic band gap of the proposed lithium niobate nanodisks in the manuscript? And please explain how the band gap suppresses the higher order modes, and then improve the color impressions.

3. Compared to the MD resonance modes in Fig.3b, the ED resonance modes are not clear.

Author Response

(The authors gave the same response as above.)

Round 2

Reviewer 1 Report

The authors addressed all my concerns, hence I support the pubblication of the manuscritp

Reviewer 3 Report

The authors addressed my comments and deleted some misleading content regarding angular dispersion effects. I have no further questions but to give my proposal of acceptance. Good luck to the authors.

Reviewer 4 Report

Since all my concerns are addressed, I think this manuscript can be accepted.